# Effect of Poly(acrylamide-acrylic acid) on the Fire Resistance and Anti-Aging Properties of Transparent Flame-Retardant Hydrogel Applied in Fireproof Glass

**DOI:** 10.3390/polym13213668

**Published:** 2021-10-25

**Authors:** Feiyue Wang, Mengtao Cai, Long Yan

**Affiliations:** Institute of Disaster Prevention Science and Safety Technology, School of Civil Engineering, Central South University, Changsha 410075, China; wfyhn@163.com (F.W.); 194812293@csu.edu.cn (M.C.)

**Keywords:** poly(acrylamide-acrylic acid), transparent flame-retardant hydrogel, light transmittance, heat insulation performance, anti-aging properties

## Abstract

Poly(acrylamide-acrylic acid) (P(AM-co-AA)) was synthesized via the copolymerization of acrylamide and acrylic acid and well characterized by Fourier transform infrared spectroscopy. Afterward, the obtained P(AM-co-AA) was blended with flame retardants to prepare transparent flame-retardant hydrogel applied in the fireproof glass. The influence of poly(acrylamide-acrylic acid) on fire resistance and anti-aging properties of the transparent flame-retardant hydrogels were studied by assorted analysis methods. The optical transparency analysis shows that the light transmittance of the transparent flame-retardant hydrogel gradually decreases with the decreasing mass ratio of acrylamide to acrylic acid in P(AM-co-AA). Heat insulation testing shows that the heat insulation performance of fireproof glass applying the transparent flame-retardant hydrogel firstly decreases and then increases with decreasing mass ratio of acrylamide to acrylic acid in P(AM-co-AA). When the mass ratio of acrylamide to acrylic acid is 1:2, the obtained P(AM-co-AA) endows the resulting flame-retardant hydrogel applied in fireproof glass with the lowest light transmittance of 81.3% and lowest backside temperature of 131.4 °C at 60 min among the samples, which is attributed to the formation of a more dense and expanded char to prevent the heat transfer during combustion, as supported by the digital photos of char residues. The results of TG analysis indicate that P(AM-co-AA) imparts high thermal stability to the resulting hydrogels due to the hydrogen bonds between carboxyl and amide groups. The accelerated aging test shows that the transparent flame-retardant hydrogel containing P(AM-co-AA) is less affected by aging conditions. Especially, when the mass ratio of acrylamide to acrylic acid in P(AM-co-AA) is 4:1, the resulting transparent flame-retardant hydrogel shows a light transmittance of 82.9% and backside temperature of 173.1 °C at 60 min after 7 aging cycles, exhibiting the best comprehensive properties among the samples.

## 1. Introduction

Fire is a major threat to business, commerce, and society, in which building fires with the highest proportion feature rapid fire development and difficult fire fighting and rescue. In particular, the multifunctionality and complexity of buildings increase the building fire risk, which calls for building fire defenses to reduce fire losses. Passive fire defenses such as fire barriers have demonstrated success in reducing fire losses by delaying or preventing products of combustion from propagating into an adjacent space. However, there are openings in many barriers permitting access for people, light, heat, air or other building services. In case of fires, these weaknesses reduce the fire resistance of fire barriers, resulting in the rapid spread of fire and toxic smoke [1,2]. Therefore, to ensure the fire resistance of fire barriers structures and buildings, the installation of fireproof glass at the openings of fire doors and windows is necessary to inhibit the entrance of fire, heat, and smoke into the adjacent spaces, according to GB50016-2018 and ISO 9051-2001 [3,4].

Fireproof glass is a special glass for maintaining the integrity and heat insulation in the prescribed fire resistance test, which can be divided into monolithic fireproof glass and composite fireproof glass with flame retardant lamination. Composite fireproof glass, which is prepared by embedding the transparent flame-retardant hydrogel interlayer between pieces of glass, has drawn more attention due to its excellent fire-resistant and heat insulation performance [5]. This interlayer is composed of inorganic and polymeric gel.

Unlike monolithic fireproof glass, when exposed to the flame, the transparent flame-retardant hydrogel interlayer in composite fireproof glass can create an expanded heat insulation char layer acting as a remarkable barrier to the spread of heat, fuel, and oxygen. The transparent flame-retardant hydrogel is usually composed of polymers and flame retardants, which is the key to the performance of the fireproof glass. Unfortunately, the poor weathering resistance of the hydrogel interlayer limits the industrial application of composite fireproof glass. The polyacrylamide hydrogel is especially susceptible to environmental factors such as ultraviolet light and temperature outdoors, leading to the deterioration of optical and fire resistance properties [6]. Therefore, there is an urgent need to improve fire resistance and anti-aging properties of transparent flame-retardant hydrogels. However, the anti-aging properties and optical transparency of P(AM-co-AA)-based transparent flame-retardant hydrogels applied in fireproof glass interlayers have not been investigated.

Hydrogels have drawn great attention as fire-resistant materials because of their high water content, water retention, temperature resistance, cooling, and inhibition properties [7]. Over the years, acrylic acid and acrylamide based hydrogels have attracted interest since they can be structurally tailored to obtain hydrogels with superior properties, including long life-to-failure duration and great water absorption capacity, etc [8,9]. Currently, many studies are devoted to the modification of hydrogels through the introduction of new monomers, metal ions, and flame retardants to enhance the fire performance of hydrogels [10]. Li et al. proposed a novel hydrogel (P(AA-co-AM)/WG gel) with excellent fire prevention and extinction performance [11]. Cheng et al. synthesized a corn straw-co-AMPS-co-AA hydrogel with good thermal stability, as well as adhesion and swelling at high temperatures [12]. Cui et al. developed Li-alginate/poly (acrylamide-co-stearyl methacrylate) hydrogels with high toughness ability and water retention capacity, which are expected to last for 6 months or even longer as fire-resistant materials [7]. Thus, it is anticipated that the P(AM-co-AA) could impart superior fire resistance and anti-aging properties to transparent flame-retardant hydrogel. On the one hand, the hydrogen bondings of the carboxyl and amide groups on the copolymer chain imparts excellent chemical and thermal stability to the hydrogel. On the other hand, the introduction of flame retardants in the polymer network can enhance the fire resistance of the transparent flame-retardant hydrogel [13]. As far as we know, there are few works reporting the effect of P(AM-co-AA) on the fire resistance and anti-aging properties of transparent flame-retardant hydrogel applied in fireproof glass.

In this paper, P(AM-co-AA) was synthesized and characterized by Fourier transform infrared spectroscopy (FTIR). Then the flame retardants were incorporated to produce transparent flame-retardant hydrogel applied in fireproof glass. The effect of the mass ratio of acrylamide to acrylic acid in P(AM-co-AA) on the optical transparency, thermal stability, heat insulation, and anti-aging properties of the transparent flame-retardant hydrogel applied in fireproof glass was assessed compared to the hydrogel without P(AM-co-AA).

## 2. Materials and Methods

### 2.1. Materials

Acrylic acid (AA, purity 99%), acrylamide (AM, purity 99%), ammonium persulfate (APS, 99.99%), *N*,*N*-methylene bisacrylamide (MBA, purity 99.5%) were obtained from Aladdin Chemistry Co., Ltd. (Shanghai, China). Sodium hydroxide (NaOH) with a purity of 98% was supplied by Chengdu Cologne Chemical Co., Ltd. Company (Chengdu, China). Sodium metabisulfite (Na_2_S_2_O_5_, 96%), ammonium dihydrogen phosphate (NH_4_)_2_HPO_4_, 99%), urea (H_2_NCONH_2_, 99%), glycerol (HOCH_2_OHCH_2_OH, 99%) were acquired from Tianjin Zhiyuan chemical reagent Co., Ltd. (Tianjin, China). Magnesium chloride hexahydrate (MgCl_2_·6H_2_O, 98%) was obtained from Sinopharm Chemical Reagent Co., Ltd. (Shanghai, China)

### 2.2. Preparation of Transparent Flame-Retardant Hydrogel

A series of transparent flame-retardant hydrogels were prepared by copolymerization of AM and AA with different mass ratios as shown in Table 1.

First, a quantity of sodium hydroxide was melted in deionized water under ice water bath conditions, and then acrylic acid was then slowly added drop-wise to the sodium hydroxide solution to obtain a sodium acrylate solution with a neutralization degree of 75%. After that, a certain amount of acrylamide, *N*,*N′*-methylene bisacrylamide and flame retardant solution were incorporated into the above solution and stirred continuously for 0.5 h at 70 °C to prepare the transparent fireproof solution. The synthesis route of the transparent flame-retardant hydrogel is presented in Figure 1.

### 2.3. Preparation of Fireproof Glass

Figure 1 presents the preparation of the fireproof glass. First, the above transparent fireproof solution was left to stand for 24 h and then filtered. Then ammonium persulfate and sodium metabisulfite were added to the mixed solution in turn while stirring constantly. This solution was poured into a 15 × 15 × 4 cm^3^ glass by siphoning at room temperature and then cured at 30–40 °C for two days to obtain the composite fireproof glass.

### 2.4. Measurements and Characterization

#### 2.4.1. Fourier Transform Infrared (FTIR) Spectroscopy

The tested hydrogel samples were kept at 50 °C for 24 h in an oven. The transparent flame-retardant hydrogels were characterized by an iCAN9 FTIR spectrometer (Tianjin Energy Spectrum Technology Co., Ltd., Tianjin, China) with KBr pellets in the wavenumber range of 4000 to 500 cm^−1^.

#### 2.4.2. Optical Transparency Analysis

Light transmittance meter LS183-108H (Shenzhen Lianhuicheng Technology Co., Ltd., Shenzhen, China) was used for optical transparency analysis. The light transmittance of the fireproof glass was tested at different locations of the sample separately, and the average value was taken. The average value of light transmittance of 10 different positions of the specimen was used to obtain the final value.

#### 2.4.3. Heat Insulation Test

The schematic of the device is shown in Figure 2. A cone heater was used to test the heat insulation performance of fireproof glass protected by transparent flame-retardant hydrogel under a heat flux of 50 kW/m^2^. The device consists of a sample table, conical heater, thermocouples, temperature recorder, etc. A 150 ×150 × 4 mm^3^ sample was placed in the center of the test table, while the K-type thermocouples placed on the back of the sample were used to record the backside temperature rise.

#### 2.4.4. Thermogravimetric Analysis

The tested hydrogel samples were kept at 50 °C for 24 h in an oven. A TGA/SD-TA851 thermogravimetric analyzer (Mettler Toledo Instruments Co., Ltd., Zurich, Switzerland) was used for TG analysis. About 3–5 mg of hydrogel was heated at a heating rate of 10 °C/min from 25 to 800 °C under a nitrogen atmosphere of 40 mL/min.

#### 2.4.5. Accelerated Aging Test

The accelerated aging test adopted a UV aging tester (Shi Haoran Machinery Equipment Factory, Dongguan, China) according to GB15763.1-2009. The light transmittance of samples was measured before exposure. The irradiation intensity of the UV lamp was 0.76 W/(m^2^·nm), and the aging cycle period was set at 24 h. The location of the samples were exposed to 1, 2, 3, 5, and 7 cycles of aging tests at 45 ± 5 °C. Samples were exchanged every 12 h during aging to ensure that the samples were exposed to the same irradiance during the same aging period.

## 3. Results and Discussion

### 3.1. Characterization of Poly (acrylic acid-co-acrylamide)

The chemical structures of P(AM-co-AA) were determined by FTIR, as shown in Figure 3. It can be seen that the broad bands around 3100–3500 cm^−1^ represent a mixed absorption stretching peak of N–H and O–H groups. The peaks at 2927 cm^−1^ (–CH_2_ groups), 1672 cm^−1^ (C=O groups), 1454 cm^−1^ (C–N groups in CONH_2_) indicate the successful synthesis of PAM. For the spectrum of P(AM-co-AA), three new feature peaks are found at 1560 and 1544 cm^−1^ (asymmetric stretching of C=O groups in COONa), 1166 cm^−1^ (stretching vibration of C–O groups), 920 cm^−1^ (deformation vibration stretching of O–H groups in dimer carboxylic acid), revealing the copolymerization between acrylamide and acrylic acid [14,15,16,17]. Moreover, the peaks of O–H groups and COONa groups strengthen gradually with the increase content of acrylic acid in P(AM-co-AA). It is confirmed by the above results that both the PAM and the P(AM-co-AA) were successfully prepared.

### 3.2. Optical Transparency Analysis

The light transmittance and digital photos of transparent flame-retardant hydrogels are shown in Figure 4.

As shown in Figure 4, the samples are completely transparent without bubbles and discoloration as seen by the naked eye. Besides, the light transmittance of flame-retardant hydrogels decreases continuously with the increase of acrylic acid content in P(AM-co-AA). When the content of the AA chain exceeded that of the AM chain, the light transmittance of C-5 decreased significantly to 81.3% and the logo under the glasses could still be clearly observed.

Generally, the optical transparency is related to the dispersion of particles in the matrix and the compatibility of the particles with the matrix [18]. The decrease of light transmittance of the hydrogel interlayer is attributed to the increase of cross-linking points and molecular weight of the copolymer, as well as the entanglement of the copolymer chains, resulting in an increase of refractive index (*n_m_*), according to the Novak equation Equation (1) [19]. The increasing *n_m_* value is ascribed to the refraction of internal flame retardant molecules which originates from the poor compatibility of acrylic acid with inorganic flame retardants [20].
(1)$$\frac{I}{I_{0}}=e^{-\left[\frac{3V_{P}xr^{3}}{4\lambda^{4}}\left(\frac{n_{p}}{n_{m}}-1\right)\right]}$$
where *n_p_* and *n_m_* are the particle refractive index and the refractive index of the hydrogel matrix, respectively [21].

### 3.3. Fire Resistance Analysis

The cone heater was used to evaluate the heat insulation of the fireproof glass applying the transparent flame-retardant hydrogel. The backside temperature curves are presented in Figure 5.

As shown in Figure 5, it is evident that the fire resistance of samples first decreases and then increases as the mass ratio of acrylamide to acrylic acid in P(AM-co-AA) decreases, indicating that fire resistance of the transparent flame-retardant hydrogels is closely related to the composition of P(AM-co-AA). In addition, the C-5 sample with a 1:2 mass ratio of acrylamide to acrylic acid demonstrates the best heat insulation property among all samples.

The backside temperature data from the heat insulation test are displayed in Table 2.

The backside temperature rise process can be divided into three stages including an initial rise stage, a stable stage, and a sudden rise stage. The transparent flame-retardant hydrogels containing P(AM-co-AA) have a slower initial temperature rise due to the enhancement of the char formation effect of the transparent flame-retardant hydrogel, as supported by the TG analysis. Compared to C-0, the samples from C-1 to C-4 reduce the backside temperature rising speed concomitant with the increasing intumescent effect and continuous denseness of the heat insulation layer, exhibiting a reduction in fire resistance capabilities. As for the C-5 sample, the heat insulation property is greatly increased compared to the C-0 sample, indicating that the introduction of high content of the AA chain in P(AM-co-AA) is beneficial for enhancing the fire resistance of the transparent flame-retardant hydrogel.

The digital photos and micrographs of the char residues obtained from the heat insulation test are shown in Figure 6. As can be seen, the char height increases with the content of the AA chain in P(AM-co-AA), and the C-5 sample exhibits the highest intumescent char layer with a height of 5.2 cm. Although the samples from C-0 to C-4 have a higher char weight, a large number of voids and cracks on the surface of the char decrease the heat insulation property of the samples. For the C-5 sample, almost no voids and cracks are found on the char surface, thus exhibiting excellent insulation property. From the top view of the char, a more continuous and dense char layer structure is formed with a decreasing mass ratio of AM to AA, which contributes to blocking the heat transfer and the escape of the pyrolytic volatiles.

The flame-retardant mechanism diagram is shown in Figure 7. The increase of fire resistance and residual weight of the C-5 sample can be explained by the fact that the presence of P(AM-co-AA) with a high content of the AA chain generates a more effective and thermally stable heat insulation barrier that prevents the transfer of internal combustible and external combustion-supporting materials. Generally, the evaporation of water molecules and decomposition of flame retardants in P(AM-co-AA) absorb a great deal of heat during the combustion process and reduce the temperature of the combustion zone, which is ascribed to the improvement of water retention and thermal stability [22].

On the basis of the above results, it can be concluded that the decomposition of ammonium dihydrogen phosphate, urea, and other flame retardants in the hydrogels produces phosphoric acid and ammonia, catalyzing the cross-linking reaction of carbon-containing compounds to form a more compact and intumescent concomitant with fewer cracks and voids, effectively segregating the heat and mass transfer to the backside of the glass during burning [23,24]. When high content AA is added, the good interaction between flame retardants and P(AM-co-AA) is benificial to form more crosslinked structures in the hydrogels that impart excellent fire resistance and water retention to the hydrogels. For example, the C-4 and C-5 samples with high AA content exert superior fire resitance and char formation among the samples. When the mass ratio of AA to AM is similar, the strong polar and hydrophilic properties of magnesium chloride will decrease the compatibility between flame retardants and P(AM-co-AA), resulting in the reduction of the flame-retardant effect of the transparent flame-retardant hydrogels.

### 3.4. Thermal Stability Analysis

The thermal stability of the transparent flame-retardant hydrogels was examined by TG/DSC test under a nitrogen environment, and the relative curves are shown in Figure 8 and Figure 9, respectively.

As can be seen from Figure 8, C-0 sample shows three decomposition stages of in the temperature ranges of 25–210, 210–350 and 350~800 °C, respectively. The first stage at around 25–210 °C is the main stage of weight loss, which is attributed to the volatilization of polymer network-bound water and the vaporization of small molecules. When the temperature increases to 210–350 °C, the adjacent amide groups in the PAM hydrogel decompose to amide groups, ammonia, and water with a slight mass loss. As the temperature rises to 350–800 °C, the PAM main chain depolymerizes to imide, nitrile, and CO_2_ with the formation of molten char, and the residue of C-0 at 800 °C is 4.3%.

As is evident from Figure 8, a higher or lower mass ratio of AM to AA in P(AM-co-AA) can increase the residual weight and water retention of the hydrogels, and the similar mass ratio of AM to AA in P(AM-co-AA) will decrease the residual weight and water retention of the hydrogel. This is especially true in the case of the C-5 sample, which has the lowest mass ratio of AM to AA and exhibits the highest residual weight of 8.7% at 800 °C. Compared to the PAM hydrogel, the smaller mass loss of the P(AM-co-AA) hydrogel in the first stage is ascribed to the fact that the volatilization of physically bound water and dehydration of hydroxyl groups is more difficult than that of free water [25]. This phenomenon can be explained by the fact that the flame retardant with strong polarity and hydrophilicity improves the degree of cross-linking of the copolymer, at the same time a large number of hydroxyl groups are introduced into the copolymer, which helps form more hydrogen bonds [19]. In addition, when the temperature increases to 350–800 °C, the decomposition of carboxyl, ammonium, and urea promote the formation of molten char and creates a high-quality char layer on the surface of the fireproof glass, thus protecting the backside fireproof glass from heat and decomposition [14,15]. In summary, the introduction of P(AM-co-AA) with an appropriate mass ratio of AM to AA could increase the water retention properties and residual weight of the hydrogels, resulting in excellent thermal stability and a char-forming effect [26].

The DSC curves of the samples are shown in Figure 9. Here, it can be seen that the DSC curves of the hydrogels show an endothermic process and an exothermic process. For the PAM hydrogel of C-0, three heat absorption peaks are found at 100–450 °C. The first endothermic peak at 150–210 °C is attributed to the decomposition of amide groups and flame retardants. The second endothermic peak at 220–300 °C is ascribed to dehydration condensation of the adjacent amide groups [17]. When the temperature increased to 300–360 °C, the PAM backbone depolymerizes to imide, nitrile, and CO_2_ with the formation of the charcoal residue. The third exothermic peak at 400–500 °C is due to the release of molecular water from the polymerization/recrystallization process. At this point, the hydrogel decomposes into CO_2_ and long-chain polymers; the reaction in this phase is still dominated by heat absorption [22]. Particularly, the C-0 sample shows the endothermic peaks at 154 and 249 °C, respectively, while C-1 sample shows the endothermic peaks at 156 and 257 °C, respectively. Compared to the C-0 sample, the C-1 sample shows a higher peak endothermic temperature due to the cyclization and dehydration of the COOH groups in the P(AM-co-AA) chain, indicating an increase of thermal stability of the C-1 sample. The exothermic process is mainly ascribed to the char formation of the hydrogel, and the C-1 and C-5 samples show a stronger exothermic process in the temperature range of 300–600 °C than that of the C-0 sample due to the better char-forming ability.

### 3.5. Anti-Aging Analysis

Figure 10 displays the digital photos and light transmittance of samples at different aging cycles. It can be seen that in the C-0 sample containing the PAM hydrogel, bubbles appear along with a turbidity phenomenon concomitant with the decrease of light transmittance after the accelerated aging test.

Compared to C-0, the optical transparency of the hydrogels containing P(AM-co-AA) gradually decrease with the decrease of mass ratio of acrylamide to acrylic acid in P (AM-co-AA). However, C-4 and C-5 samples show discoloration in some areas after 7 aging cycles, indicating that the high proportion of acrylic acid has a negative effect on the transparency of the P (AM-co-AA) hydrogel. The turbidity phenomenon is attributed to the precipitation of flame retardants and the decomposition of the transparent flame-retardant hydrogel, and the bubbles originate from the voids created by the hydrogel volume contraction under UV exposure. The discoloration of C-4 and C-5 mainly results from the chromophores produced by UV radiation [25].

Figure 11 illustrates the light transmittance decreases of samples after 7 aging cycles. It is evident in Figure 11 that the light transmittance of the C-0 sample decreases significantly up to 25.1% after 7 aging cycles compared to that of the samples containing P (AM-co-AA). In particular, the light transmittance of C-1 sample decreases by only 2.9% when the mass ratio of acrylamide to acrylic acid is 4:1 in P (AM-co-AA), indicating that the presence of P (AM-co-AA) with a low content of AA chain could effectively enhance the anti-aging properties of the hydrogels.

The backside temperature curves of the samples after 7 aging cycles are illustrated in Figure 12, and the back temperature data is given in Table 2.

As shown in Figure 12, compared to unaged samples, the time of the initial stage is significantly shorter and the stable temperature increases by about 20 °C, which can be attributed to the deterioration of hydrogel stability caused by the degradation of the polymer chains during aging treatment. Compared to C-0, the transparent flame-retardant hydrogels containing P(AM-co-AA) exhibit a longer initial temperature rise stage, a lower stable temperature, and small changes in fire resistance after 7 aging cycles, indicating an enhancement in anti-aging properties.

As seen in Table 2, the backside temperature changes before and after the aging test at 3600 s, increasing with the decrease of mass ratio of acrylamide to acrylic acid in P (AM-co-AA), indicating that the introduction of P (AM-co-AA) containing a high content of AA chain can more effectively improve the fire resistance and anti-aging properties of the transparent flame-retardant hydrogel. In addition, when the mass ratio of AM to AA is 4:1, the resulting P(AM-co-AA) shows the best anti-aging properties to the transparent flame-retardant hydrogel.

In order to further investigate the anti-aging mechanism of hydrogel, the unaged hydrogels, and aged hydrogels of C-0 and C-1 with 0, 1, 3, and 7 aging cycles were examined by FTIR analysis, as shown in Figure 13a,b, respectively. For the PAM hydrogel, the specific changes of characteristic peak intensity of aged and unaged samples are given in Table 3.

As shown in Figure 13a, the stretching vibration peak of C=O groups becomes stronger with the increasing aging cycles, which corresponds to the UV radiation products of the aging process, leading to a reduction in the light transmittance of the hydrogel [27]. The –CH_2_ stretching vibration peak at 2920 cm^−1^ becomes weaker as the aging cycle increases, corresponding to the degradation process of the polymer chain.

As shown in Figure 13b, compared to the PAM hydrogel of sample C-0, the intensity of the C=O group at 1672 cm^−1^ in the C-1 sample becomes slightly stronger with the increase of aging cycles because the high stablity of carboxylates and carboxylic acids [28], indicating the excellent anti-aging properties of the C-1 sample. The stretching vibration peak of NH_2_ at about 1400 cm^−1^ first increases and then decreases during the aging test, which corresponds to the degradation of the polymer chain and the decomposition of the polyacrylamide respectively. The peak intensities of O–P=O and P–O groups at about 1300 and 900 cm^−1^ gradually increase during the aging test, which is ascribed to the precipitation of the flame retardants. The stretching vibration peak of C–O groups at 1150 cm^−1^ in P(AM-co-AA) hydrogels gradually strengthens with the increase of aging cycles, which is consistent with the trend of heat insulation performance and light transmittance.

## 4. Conclusions

In this study, acrylamide and acrylic acid were selected as the copolymer monomers ammonium persulfate and sodium metabisulfite as the initiators, *N,N′*-methylenebisacrylamide as the crosslinker to prepare a series of P(AM-co-AA) with different compositions via the solution polymerization. Then, magnesium chloride, urea, and other flame retardants were incorporated into the obtained P(AM-co-AA) to prepare transparent flame-retardant hydrogels applied in fireproof glass. The results reveal that the light transmittance of the hydrogel decreases continuously with the increase of acrylic acid fragments in P(AM-co-AA). Especially, when the mass ratio of AM to AA is 1:2, the obtained P(AM-co-AA) imparts the lowest light transmittance of 81.3% to the resulting C-5 sample. The results of the fire resistance analysis illustrate that the heat insulation performance of the transparent flame-retardant hydrogel applied in fireproof glass firstly decreases and then increases with the decreasing mass ratio of acrylamide to acrylic acid in P(AM-co-AA), and a low content of AA chain in P(AM-co-AA) will diminish the flame-retardant effect of the transparent fire-retardant hydrogel. When the mass ratio of AM to AA in P(AM-co-AA) is 1:2, the resulting C-5 sample shows a supperior flame-retardant effect, which shows a 16.8% reduction in equilibrium backside temperature at 60 min compared to the C-0 sample containing PAM. In addition, the results of TG analysis demonstrate that the presence of P(AM-co-AA) enhances the thermal stability and char-forming capabilities of the hydrogels owing to the stable hydrogen bonding and cyclization reaction between carboxyl; and the C-5 sample acquires the highest residual weight at 800 °C of 8.72% among these samples. The superior heat insulation performance of the C-5 sample is attributed to the excellent water retention properties and the formation of more thermally stable cross-linking structures and aromatic structures that generate a high-quality char layer to prevent heat transfer, as determined from the results of char residue analysis. The results of the accelerated aging test suggest that the introduction of P(AM-co-AA) could significantly improve the anti-aging properties of the transparent flame-retardant hydrogel, thus providing superior durability of the fireproof glass applying the hydrogel. Especially, C-1 shows the light transmittance of 82.9% and equilibrium backside temperature of 173.1 °C at 60 min after 7 aging cycles, exhibiting the best comprehensive properties among the samples. In general, AA is an effective synergist in preparing the transparent flame-retardant hydrogels with excellent fire resistance and anti-aging properties.

## Figures and Tables

**Figure 1 polymers-13-03668-f001:**
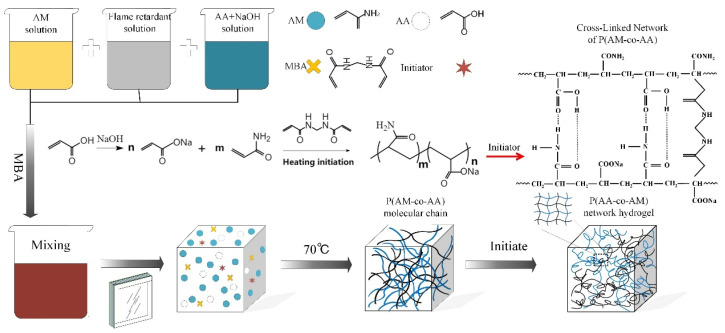
Preparation process of P(AM-co-AA) hydrogel.

**Figure 2 polymers-13-03668-f002:**
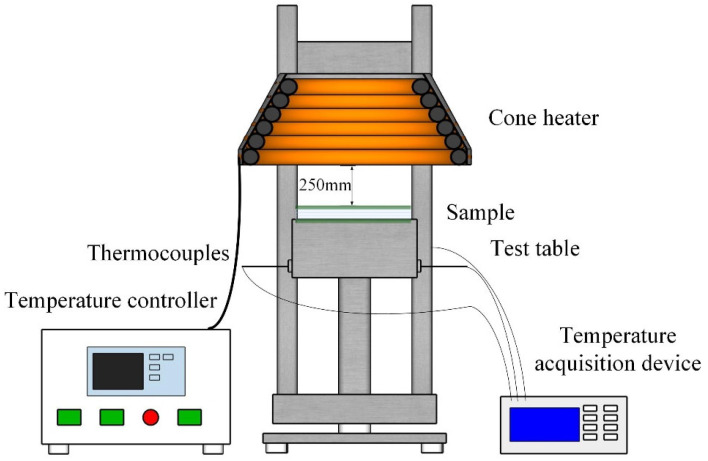
Schematic diagram of heat insulation test.

**Figure 3 polymers-13-03668-f003:**
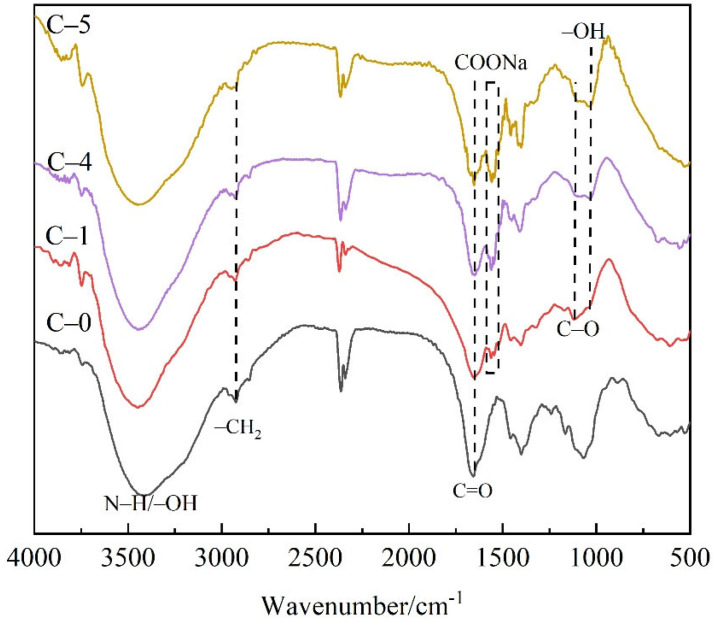
FTIR spectra of transparent flame-retardant hydrogels.

**Figure 4 polymers-13-03668-f004:**
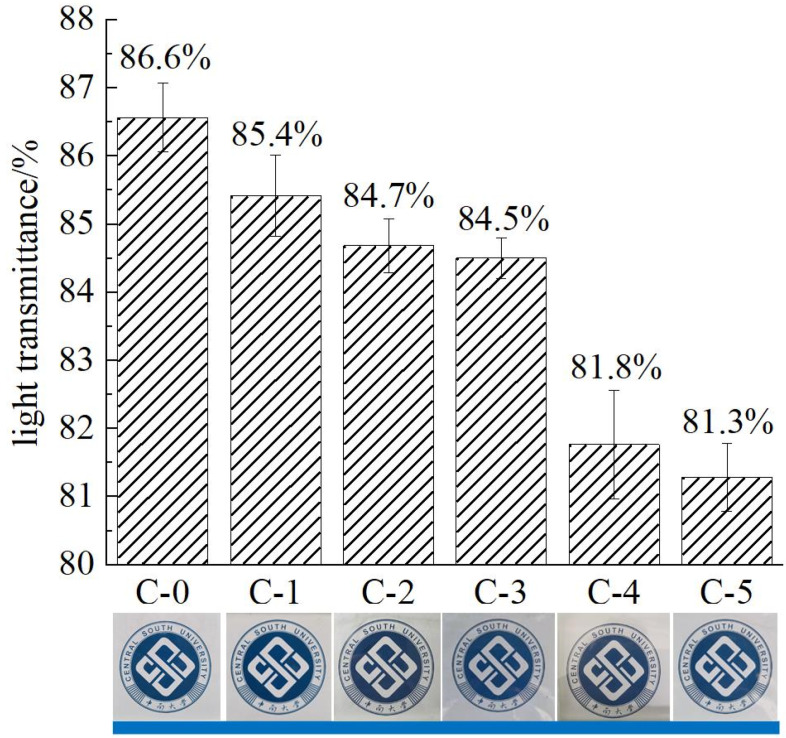
Light transmittance of fireproof glass samples.

**Figure 5 polymers-13-03668-f005:**
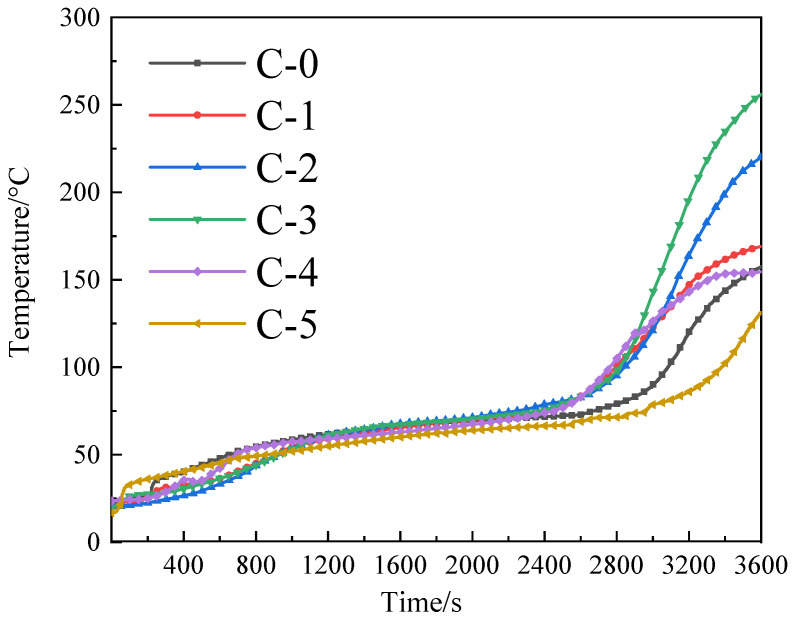
Backside temperature curves of the transparent flame-retardant hydrogels.

**Figure 6 polymers-13-03668-f006:**
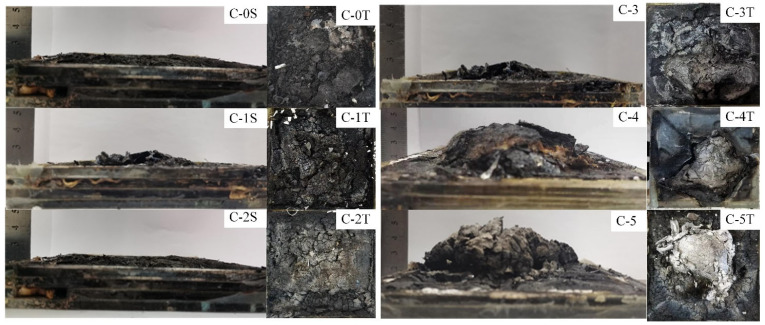
The digital photos of the transparent flame-retardant hydrogels after the heat insulation test (Note: letter S presents the side view of the char, letter T presents the top view of the char).

**Figure 7 polymers-13-03668-f007:**
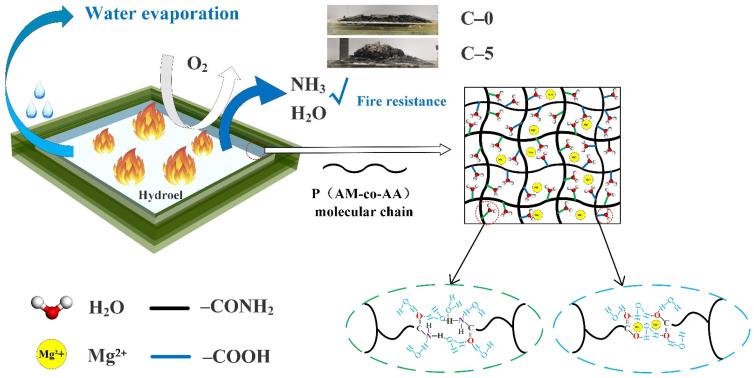
Fire retardant mechanism of P(AM-co-AA) hydrogel in fireproof glass.

**Figure 8 polymers-13-03668-f008:**
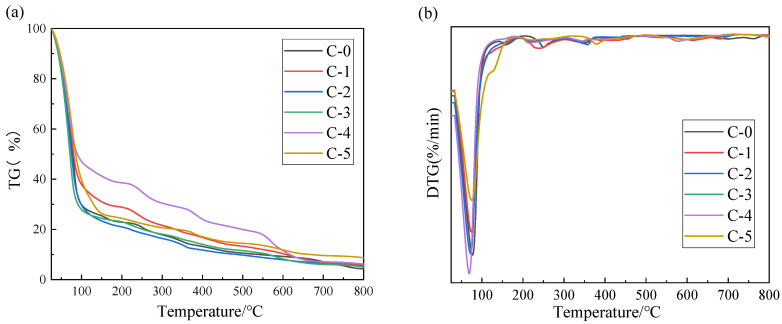
TG (**a**) and DTG (**b**) curves of the Samples.

**Figure 9 polymers-13-03668-f009:**
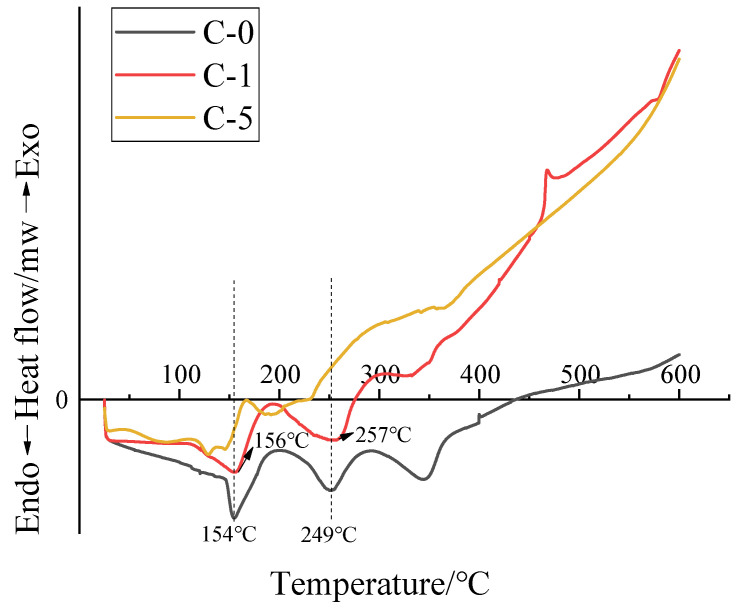
DSC curves of the Samples.

**Figure 10 polymers-13-03668-f010:**
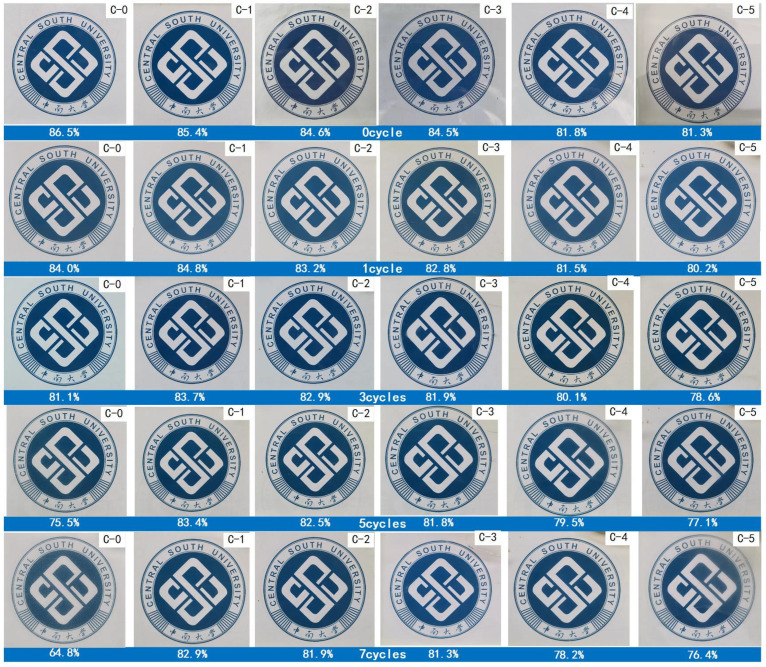
Digital photos and light transmittance of samples at different aging cycles.

**Figure 11 polymers-13-03668-f011:**
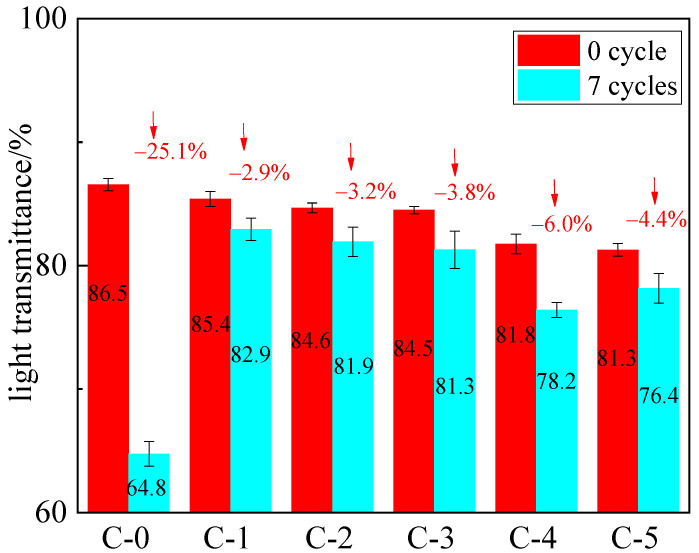
Light transmittance of the flame-retardant hydrogel before and after 7 aging cycles.

**Figure 12 polymers-13-03668-f012:**
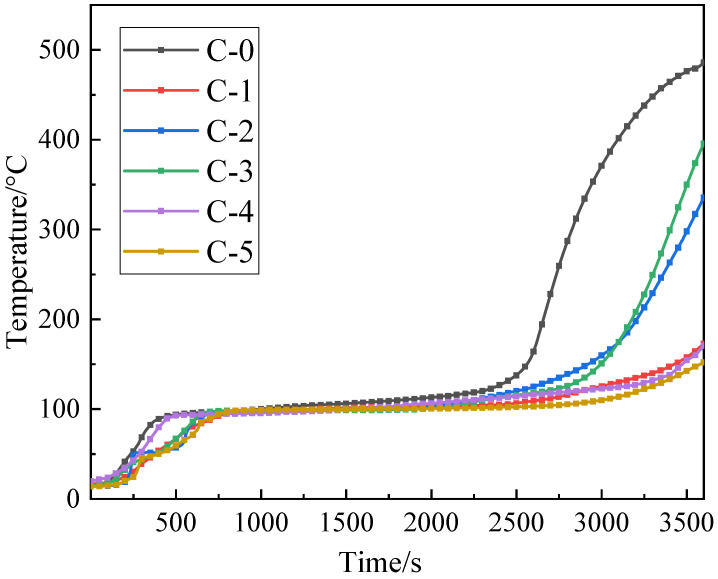
Backside temperature curves of aged samples.

**Figure 13 polymers-13-03668-f013:**
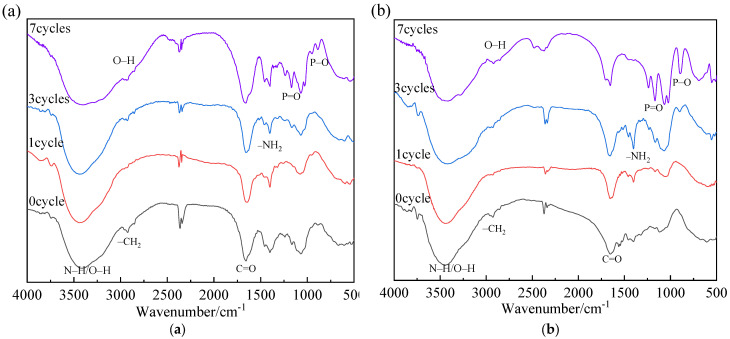
FTIR spectra of C-0 sample (**a**) and C-1 sample (**b**) after different aging cycles.

**Table 1 polymers-13-03668-t001:** Compositions of the transparent flame-retardant hydrogels (mass fraction) %.

Samples	AM/%	AA/%	MBA/%	Initiator/%	Flame Retardants /%	Deionized Water/%
C-0	9	0	0.075	0.09	13	78
C-1	7.2	1.8	0.075	0.09	13	78
C-2	6.75	2.25	0.075	0.09	13	78
C-3	6	3	0.075	0.09	13	78
C-4	4.5	4.5	0.075	0.09	13	78
C-5	3	6	0.075	0.09	13	78

**Table 2 polymers-13-03668-t002:** Backside temperature values of the hydrogels before and after aging test.

Samples	Before Aging Test		After Aging Test			
	Temperature at 3600 s	Temperature Range in Stable Stage	Temperature at 3600 s	Changes	Temperature Range in Stable Stage	Changes
C-0	158.0 °C	80–100 °C	485.9 °C	327.9 °C	100–120 °C	Increased 20 °C
C-1	169.1 °C		173.1 °C	4 °C		
C-2	220.1 °C		335.4 °C	115.4 °C		
C-3	255.2 °C		395.7 °C	140.7 °C		
C-4	154.5 °C		170.6 °C	16.1 °C		
C-5	131.4 °C		152.0 °C	20.6 °C		

**Table 3 polymers-13-03668-t003:** FTIR assignments for the functional groups of C-0 and C-1 after aging test.

IR Band cm^−^^1^	Functional Groups	After 7 Cycles	Observations	
			Intensity	Changes
2920	C–H stretching in alkane	−	Weak	Significantly decreased
1672	C=O stretching in acrylamide	+	Strong	Significantly Increased
1150	C–O stretching in ester group	+	Strong	Slightly Increased

Note: “−” represents a decrease in peak intensity, “+” represents peak intensity enhancement.

## Data Availability

The data presented in this study are available upon request from the corresponding author.

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
