# Peer review of "Effect of Poly(acrylamide-acrylic acid) on the Fire Resistance and Anti-Aging Properties of Transparent Flame-Retardant Hydrogel Applied in Fireproof Glass"

_polymers, 2021, doi:10.3390/polym13213668_

Round 1

Reviewer 1 Report

This paper deals on the fire resistance of fireproof glasses, FPGs, by a careful selection of the hydrogel to be sandwiched between glasses in composites FPGs. This topic is of extreme relevance and can be of interest for many readers.

Tests on effectiveness of the fire protection, transparency and ageing resistance have been carried out on several FPGs in which in the hydrogel the relative amount of Acrylamide/Acrylic acid has been changed.

While the research design can be shareable, results and discussion must be strongly improved.

Copolymer composition.

In fig 1 an alternate AM-AA copolymer is shown, with secondary bonds between acrylate and amide put in evidence. However, in fig 7 two homopolymer PAA and PAM are shown. The monomers sequences are driven by the reactivity ratio and monomer feed composition (which generally does not corresponds with the copolymer composition): optimal reactivity ratios are rAAm1.33 and rAAc 0.23 in a basic environment, which predict a copolymer with longer AM sequences and very shorter sequences of AA. Any discussion about intraction between chain must take in account these structures.

IR analysis

First, the fire retardant system used in the hydrogel (13%) is not described in the section materials. Only in the conclusions some indication is suggested about (MgCl2 (?) urea and APP?). It makes the infrared analysis and attribution of the bands doubtful and inaccurate.

Urea and APP exhibit absorption in the 1700-1600 cm-1 and 1300-1000 cm-1. 

In the hydrogel the amount of water is quite high, as a consequence the broad bands around 3100-3500 cm−1 and 1630 cm-1  should be attributed mainly to water. Acrylamide does not possess -CH3 groups therefore the absorption at 2927 should be due to other species (maybe impurities)

Notably, in fig 3 the attribution of 2200-2300 cm-1 bands to -NH is wrong (it is CO2). The same in fig. 13.

Fire resistance analysis

Any mechanism proposed must take in account the kind of fire retardant used, however presently there is not enough information about this point, as said before

Thermal analysis

“The water retention properties and residual weight of the P(AM-co-AA) hydrogels gradually increases with the increasing AA content in P(AM-co-AA)” This is not what appears in fig 8.

DSC thermogram are hardly interpretable

Reviewer 2 Report

Review of article: ,, Effect of poly (acrylamide-acrylic acid) on the fire resistance

and anti-aging properties of transparent flame-retarded hydrogel applied in fireproof glass’’.

  1. ,,backside temperature.’’ What i sit? What does it mean?
  2. Introduction:…which are expected to last for 6 months or even longer as fire-resistant materials[7]-why so short?
  3. In table 1 : 7.2 -the numbers fell apart
  4. Point 2.4.5, incomprehensible sentence: ,,Samples were exchanged every 12 hours during aging to ensure that the samples were exposed to the same irradiance during the same aging period.’’
  5. Point 3.2. by what method (how) was transparency tested?
  6. Point 3.3. There is no table 2.
  7. Please explain ,, back temperature’’
  8. Thermal analysis was performed. The focus was on the analysis of transparency. There is talk of fire resistance. However, fire resistance has not been demonstrated. No flammability tests. Only showing pictures of more or less charred samples is not a fire resistance analysis.
  9. It is not entirely clear (not explained in the introduction) whether P (AM-co-AA)) is glass or an additive to glass or a hydrogel (additive to glass).
  10. Point 2.3. Glass, before being made fireproof, was called fireproof. Is it right?
  11. Point 3.3. Insulation analysis mixed with fire resistance.

Reviewer 3 Report

The manuscript under the title: “Effect of poly (acrylamide-acrylic acid) on the fire resistance and anti-aging properties of transparent flame-retarded hydro-gel applied in fireproof glass” is in line with Polymers journal. It based on original research. The article has a typical organization for research articles. Before the publication it requires significant improvements, especially:
1.    Introduction: it is necessary to clearly formulate the scientific novelty of the research and show how it differs from those known in the literature.
2.    From the described method of preparation of fireproof glass, it is not clear how the authors avoided the formation of air bubbles during pouring. This needs to be described in detail.
3.    SEM images of char samples after heat insulation test of flame-retarded hydrogels should be provided and discussed.
4.    When describing the mechanism for reducing the combustibility of samples, the related references should be cited corresponding to each aspect, e.g. (but not limited to these): Inorg. Mater. Appl. Res. 2019, 10, 1135–1139, https://doi.org/10.1134/S2075113319050228; AIP Conference Proceedings 1899, 020003 (2017), https://doi.org/10.1063/1.5009828.

5.    Discussion: please compare achieved results with up-to-date literature, also with composites with other admixtures. Discuss the achieved results.

Round 2

Reviewer 1 Report

After correction this paper has been improved. Due to the interest of the topic, I think that this paper can now be published if some points will be further clarified

Thermal analysis

I still think that the sentence “The water retention properties and residual weight of the P(AM-co-AA) hydrogels gradually increases with the increasing AA content in P(AM-co-AA)” does not represent what in fig 8.

AA content is in the order C5>C4>C3>C2>C1

Water retention order:C4<C1<C5<C0<C3<C2

Residual weight (800 °C) order C5>>C4>C1>C2=C3>C0

Thus C1, that has the lower amount of AA in copolymers, exhibits good performance bot in residual weight and in water retention

DSC thermograms are very hardly interpretable. I suggest the author to do again these tests. In addition, in fig 9 the y axis label seems to be wrong (exotherm and endotherm are exchanged?)

IR analysis

In fig 13 there is no comment on the band at about 1300 cm-1 (P=O in orthophosphoric acids?) which first increases and then decreases during aging.

In conclusion C5, C4 (Higher amount of AA) and C1 (lower amount of AA) exhibit good properties bot in fire resistance in transparency and in ageing resistance. C2 and C3 which have an intermediate amount of AA perform much worse. Why?

Eventually please correct typing mistake:  thermal analysis equipment is from Mettler Toledo

Reviewer 2 Report

The article has been greatly improved. I request its publication. 

Author Response

Thank you for your approval. We have optimized the manuscript according to the opinions of other reviewers.

Reviewer 3 Report

The authors edited the manuscript well; all the recommendations of the reviewer were taken into account. I recommend this article for publication in the journal "Polymers"

Author Response

(The authors gave the same response as above.)

Round 3

Reviewer 1 Report

After the last extensive revision I think that this paer can now be published